# High-Resolution Estimation of Suspended Solids and Particulate Phosphorus Using Acoustic Devices in a Hydrologically Managed Canal in South Florida, USA

**DOI:** 10.3390/s23042281

**Published:** 2023-02-17

**Authors:** Ikechukwu S. Onwuka, Leonard J. Scinto, David C. Fugate

**Affiliations:** 1Department of Earth and Environment, Florida International University, Miami, FL 33199, USA; 2Institute of Environment, Florida International University, Miami, FL 33199, USA; 3Department of Marine and Earth Sciences, Florida Gulf Coast University, 10501 FGCU Blvd. South, Fort Myers, FL 33965, USA

**Keywords:** acoustic devices, surrogate methods, high-resolution estimates, suspended solids, particulate phosphorus, canals

## Abstract

Conventional methods of measuring total suspended sediments (TSS) and total particulate phosphorus (TPP) are typically low-resolution and miss critical processes that impact their exports in aquatic environments. To create high-resolution TSS and TPP estimates, echo intensity (EI), a biproduct of velocity measurements from acoustic devices, was utilized. An acoustic Doppler velocimeter (ADV) and an acoustic Doppler current profiler (ADCP) were deployed in three locations in the L-29 Canal in South Florida, USA, to obtain estimates near the canal bed and in the water column, respectively. Corrections for transmission losses from the ADCP proved unnecessary due to the low vertical variability in the measured EI. EI calibrations were performed using artificially created TSS obtained from bed sediments (ADV) and gravimetrically measured TSS from water samples that matched the depths and times of the ADCP deployments. The measured TSS values were then analyzed for total phosphorus and converted to TPP estimates. The results showed that high TSS and TPP were caused by the rapid discharge releases typical of managed canals. This work demonstrates that high-resolution estimates are imperative for assessing the effects of such swift hydrologic changes on the potential export of sediments and nutrients to delicate ecosystems downstream.

## 1. Introduction

The movement of suspended sediments impacts the quality of aquatic systems, including freshwater environments, because they transport particulate nutrients [1,2,3,4]. Phosphorus (P) is one such nutrient [5,6,7], and excess P is responsible for adverse water quality effects, such as eutrophication and harmful algal blooms [8,9]. Due to hydrologic and biogeochemical processes, suspended sediments and their associated nutrients are dynamic in space and time. Therefore, high-resolution measurements are necessary to fully evaluate the conditions that can increase sediment and nutrient loadings to enable the restoration and preservation of impaired aquatic systems [3,10,11,12].

Total suspended sediment (TSS) concentrations are generally determined in two ways, namely through direct and surrogate methods. Direct methods involve the conventional collection of water samples via bottles and pumps, which are then analyzed gravimetrically in the laboratory [10,13,14]. This method is usually expensive and labor-intensive, often requiring costly field sampling techniques and laboratory protocols; intrusive, as delicate flocculated aggregates can easily be broken down during sample retrieval or handling; and, in some instances, sampling is physically impossible or dangerous for field personnel [15,16,17]. Furthermore, the collected samples may not be representative of the range of conditions in aquatic systems due to the low temporal and spatial resolutions of the sampling regimes employed [3,13,16].

Surrogate methods are indirect ways of estimating TSS concentrations without a reliance on water sample collection and can thus be implemented autonomously [13]. Examples of surrogate methods include optical backscatter sensors, transmissometers, laser diffraction, and acoustic devices, which all make nonintrusive high-temporal-resolution estimates of TSS [10,13,18]. Acoustic devices are resistant to biofouling, unlike optical backscatter sensors [19,20], and can also make high-resolution spatial measurements [13,17].

Commonly used acoustic devices include acoustic Doppler velocimeters (ADVs) and acoustic Doppler current profilers (ADCPs). These devices are primarily velocity instruments that measure the frequency shift (Doppler shift) of transmitted sound waves (back) scattered by suspended materials in the water and convert the sound waves or echoes to three-dimensional (north/south, east/west, and vertical) flow velocities [10,13,21]. Acoustic Doppler velocimeters (ADVs) make single-point velocity measurements for a small water volume (measurement volume) at a point approximately 5–10 cm away from the device [13]. Acoustic Doppler current profilers (ADCPs) can further provide velocity by splitting reflected sound signals into segments (range gating) so that velocities of water current can be determined at preset or user-predetermined intervals along the acoustic path (called cells or bins) [10]. ADCPs have the added advantage of providing data that reflect conditions across large areas in the water column [13].

Backscattered sound waves (acoustic backscatter) from ADV and ADCP velocity measurements have been used to estimate TSS in aquatic systems [10,14,18,22,23,24,25,26]. The fundamental theory is that acoustic waves moving through water containing sediments will scatter and attenuate depending on the characteristics of the fluid, sediment (concentration, shape size), and acoustic device [25]. Furthermore, adjustments (corrections) are usually introduced to account for the influence of the fluid and device characteristics [13,25]. Estimates of TSS are obtained by calibrating the acoustic backscatter with actual measured TSS from the deployment site using regressions [10,23]. This calibration can then be used to provide a time series of TSS estimates during the deployment of the acoustic sensor at the site.

Although high-resolution estimates of TSS now exist, there is still a need to develop similar estimates for sediment-associated constituents such as particulate phosphorus to monitor and prevent excessive nutrient loading into delicate ecosystems. Studies have found higher concentrations of TSS and phosphorus during high discharge events in rivers and canals, with sediment resuspension and desorption of sediment P highlighted as potential reasons for such high concentrations [27,28].

The human-made canals of South Florida (home of the Everglades) were constructed to meet the agricultural, urban, and environmental needs of the region [29,30]. Discharges in these canals are highly managed through inflow structures [30], and it is not uncommon to have large masses of water discharged into canals over relatively short periods. Therefore, there is a need to understand how rapid discharge releases can impact the export of sediments and phosphorus into the receiving oligotrophic Everglades wetlands. The objectives of this study were to (1) use acoustic backscatter to develop high-resolution TSS and TPP estimates in a major South Florida canal and (2) evaluate how canal discharges can impact the downstream export of TSS and TPP.

## 2. Materials and Methods

### 2.1. Instrumentation

#### 2.1.1. Characteristics and Limitations of Acoustic Devices

Acoustic devices collect many types of data, including velocity, signal amplitude/signal-to-noise ratio (SNR), correlation coefficient, temperature, and pressure (Table 1). Velocity is typically the primary parameter of interest, while the signal amplitude and correlation are used to provide data quality information [31]. The main purpose of the signal amplitude is to determine if there are enough particles in the water, making it an excellent illustration of sediment dynamics and fluctuations [32]. This signal amplitude constitutes the acoustic backscatter parameter used as a surrogate for total suspended solids. The signal amplitude data are usually accessed from the device as either signal amplitude (units of counts) or the SNR. For SONTEK instruments, the SNR is obtained from the signal amplitude by subtracting the ambient, background electronic noise level and converting it to decibel units (dB) (multiplying by 0.43) [32]. To obtain accurate velocity measurements, SNR values must be at least 5 dB (ADV) and 3 dB (ADCP) (Table 1). An SNR value of 0 dB indicates that the water is too clear and that there is no distinction between the signal (acoustic energy) and the ambient noise level [31,32]. The ADV uses the correlation coefficient as a second data quality parameter, and it is expressed as a percentage. A correlation score of 100% means reliable low-noise velocity measurements, which occur at an SNR level of about 15 dB, while a 0% correlation reflects velocity measurements dominated by noise [31,32]. Interferences that can lead to low correlation values include high velocities, turbulence, and aerated water. A score of 70% or higher indicates accurate velocity data [32].

#### 2.1.2. Acoustic Method

The conversion of the acoustic backscatter to TSS estimates (TSS) requires a series of steps that are summarized in this equation:(1)$$\mathrm{TSS}=10^{\left[A+B*\mathrm{RB}\right]}$$
where A—intercept, B—slope, and RB—relative acoustic backscatter.

The exponent of Equation (1) contains a term for the relative acoustic backscatter, RB, measured by the acoustic device, and the terms for the equation coefficients—intercept, A, and slope, B. These coefficients are determined by regressions of backscatter with artificially created TSS concentrations or gravimetrically (differences in weights) measured TSS from water samples [10]. This equation can also be expressed as:(2)Log TSS=A+B∗RB

RB is composed of the measured acoustic backscatter and a correction for transmission losses [22] in units of counts or decibels (dB), as shown below:(3) RB=RL+2TL
where RL is the reverberation level (the measured backscatter) and 2TL is the two-way transmission loss, assumed to be the same in each direction. The RL can be used directly as the actual reported values from the device (or the reported values minus the noise level) if the noise level is relatively constant [13]. This RL is multiplied by a manufacturer-supplied scale factor to convert it from count to decibels (dB). After the conversion, the RL can be referred to as the echo intensity (EI).

The two-way transmission loss, 2TL, is introduced to compensate for losses due to the spherical spreading of the acoustic beam from the acoustic device and losses due to absorption in the water, as shown below:(4)$$2\mathrm{TL}\;=20{\;\log}_{10}\left[R\;\Psi \right]+2\alpha R$$
where R—Range, Ψ—near-field correction, and α—absorption correction.

The first part of the equation is the correction for beam spreading, and the second part is the absorption correction. R is the slant range from the transducer head (ADCP) to the measured bin (m) given as:(5)$$R=r+\frac{D}{4}$$
where r is the slant distance from the transducer head to the center of the bin in meters and D is the bin size in meters [24]. The next term, Ψ, is the transducer near-field correction [37] that accounts for the non-spherical spreading of acoustic energy close to the transducer and is defined as:(6)$$\Psi \;=\left[1+1.35z+{\left(2.5z\right)}^{3.2}]/[1.35z+{\left(2.5z\right)}^{3.2}\right]$$
and
(7)$$z=\frac{R*\;\lambda \;}{{\pi *a}_{t}^{2}}$$
where a_t_ is the transducer radius in meters and λ is the acoustic wavelength in meters. However, Ψ is empirically derived and generally not as reliable as the other parts of the transmission loss equation, and could, thus, introduce some level of uncertainty [25].

The next term in the 2TL equation is the α coefficient, which describes the absorption of energy by water (α_w_) and attenuation due to suspended sediments (α_s_)—that is, α = α_w_ + α_s_ (all in dB m^−1^). The coefficient α_w_ describes the absorption of energy by water and depends on the salinity, temperature, and pressure of the water and the sound frequency of the acoustic device [13,25]. Pressure does not have a significant effect for shallow water environments (depth ≤ 20 m), and salinity is not applicable in freshwater systems [13,24], leaving temperature and frequency as the determining variables in canals, rivers, and lakes. Next, the coefficient α_s_, the sediment attenuation coefficient, is controlled by the sediment characteristics for a given acoustic frequency and is negligible in water with a TSS concentration less than 300 mg L^−1^ [38].

The correction for transmission losses is more significant for ADCPs, where the range, R, can extend many meters, while ADVs have much shorter ranges (e.g., 10 cm), and any losses are comparatively negligible [13].

### 2.2. Study Area

The Tamiami (L-29) Canal in the Everglades region in South Florida was created by means of the excavation of limestone used to construct the Tamiami Trail roadway (Figure 1). The roadway acted as a barrier to natural overland flow and restricted freshwater inflows to the Northeast Shark River Slough (NESRS) within the Everglades National Park (ENP). To remediate the limitations on freshwater inflows, the Modified Deliveries Project was created to develop the necessary infrastructure to allow more freshwater delivery into the ENP and restore a “more natural” hydrologic condition [39,40]. The infrastructure includes the construction of approximately 5.8 km of bridges on the Tamiami Trail to enable more flow into the NESRS through the S333 water control (inflow) structure (spillway) along L-29 [41]. A section of the L-29 Canal east of the spillway was used in this study.

### 2.3. Deployment in the Tamiami (L-29) Canal and Backscatter Processing

Two acoustic devices, a 10 MHz side-looking ADV and a 1500 kHz Argonaut ADCP, both manufactured by SonTek (SonTek—a Xylem Brand, San Diego, CA, USA), were deployed at three sites in the Tamiami (L-29) Canal, namely upstream (UP—25°45′40.8″ N, 80°40′18.8″ W), downstream (DS—25°45′40.6″ N, 80°39′06.3″ W), and interior (INT—25°45′39.1″ N, 80°32′03.5″ W), at different periods between June and December 2021. The ADV and ADCP were deployed during the same period at DS (June) and INT (August), while at UP, the ADV was deployed in November, and the ADCP was deployed in September. UP was closest to the inflow structure S333, while DS and INT were located before the 4.2 km and 1.6 km bridges, respectively. These sites were strategically chosen to determine how discharge releases into the L-29 Canal can transport suspended sediments and particulate phosphorus into the ENP via the bridges.

#### 2.3.1. Acoustic Doppler Velocimeter (ADV)

The ADV was deployed near the canal bed (0.14 m–0.5 m above the bed) to measure the velocity and backscatter every 2 h. The ADV had three receivers and made measurements 10 cm away from the probe (sampling volume). The backscatter from the three receivers was averaged and converted from signal amplitude (counts) into echo intensity (decibels) by multiplying by a 0.43 scaling factor [32] before being used for calibration.

#### 2.3.2. Acoustic Doppler Current Profiler (ADCP)

The three-beamed Argonaut ADCP was deployed at the canal bottom, in an upward facing direction, and was configured to make temporal and vertical velocity and backscatter measurements with a vertical resolution (bin size) that was either set at 0.5 m or 1 m (Figure 2). During deployments, the water depths in the canal ranged from 3.6 to 5.0 m and provided between 4 and 7 measurement bins. During longer deployments (1–3 weeks), the ADCP was programmed to make either 120- or 300-s measurements (averaging interval) every 1200 or 3000 s, respectively (sampling interval). For shorter deployments, the ADCP was programmed to make 10-s measurements every 100 s. Additionally, the blanking distance—the region directly above the ADCP where no measurements can be made—ranged from 0.5 m to 1 m.

### 2.4. Acoustic Backscatter Conversion and Correction

For the ADV, laboratory experiments were conducted to develop calibration curves between TSS and echo intensity (EI). Sediment cores from the surface of the canal bed were first taken from each site. Then, portions of the cores were weighed (wet weight), dried to a constant weight and reweighed to determine the water content of the sediments. Next, more fresh wet surface sediments were weighed, and the water content was accounted for so that the true (dry) weights of the sediments could be calculated. These true weights were then diluted in measured volumes of tap water in a bucket to artificially create TSS between 5 mg L^−1^ and 150 mg L^−1^ in a large bucket. Next, a pump was inserted into the bucket to create velocity and keep the sediments in suspension [26]. During agitation, the ADV was introduced into the bucket and programmed to make 15-s velocity and backscatter measurements for approximately 2 min. The agitation and measurements were repeated for all the created TSS (between 5 mg L^−1^ and 150 mg L^−1^), and the corresponding acoustic backscatter from the ADV was recorded, converted to echo intensity (EI) by multiplying by 0.43, and used to generate a calibration curve for each site.

To develop time series profiles of suspended sediments from ADCP backscatter, a series of steps were followed. First, the backscatter was downloaded from the ADCP and averaged (the three beams of the ADCP produced three backscatter values per measurement). In cases when the backscatter from one beam was much higher than the other, or when the difference in SNR values was greater than 10 dB [42], the averages of the other two beams were used. Next, the backscatter was converted to EI in dB at each bin. Since the noise level was relatively constant during deployment, it was not subtracted from the backscatter before conversion to EI. Then, the EI was corrected for transmission loss from beam spreading and attenuation due to water to give the final RB (Equation (3)). The α*_w_* values were based on the instrument frequency and water temperature since the study site was a freshwater canal with a water depth much less than 20 m. Published α*_w_* values provided by [25] were used. Attenuation due to sediment was ignored because of the low TSS measured in the water column. Data and graphical analyses were conducted using R [43], Microsoft Excel and PowerPoint, and MATLAB [44]. An alpha (*p*) value to define statistical significance was set at 0.05 for calibration regressions.

### 2.5. Water Sampling for TSS and TPP for Calibrations and Estimations

To provide TSS measurements for the calibration of the ADCP backscatter, water samples were collected simultaneously during the ADCP deployments (at UP, a calibration performed in November was applied to the September deployment). To perform the calibrations, water samples were collected from different depths in the water column that matched the bins of the ADCP (Figure 2). A peristaltic pump was used to collect the water samples in prerinsed 1 L sampling bottles, which were kept on ice and transported to the lab where they were analyzed gravimetrically for TSS. In the laboratory, predried (105 °C for 1 h) and preweighed 47 mm GFF Whatman filters were used to filter 1 L (volume) of the collected water samples (Wpre). After filtration, the filter and particles (residue) were dried again (105 °C for 1 h), cooled in a desiccator, and weighed (Wpost) [14,45]. TSS was then calculated according to the following equation:(8)$$\frac{\mathrm{TSS}\;\left({\mathrm{mgL}}^{-1}\right)=\mathrm{Wpost}\left(g\right)-\mathrm{Wpre}\;\left(g\right)*1000\;}{V\left(L\right)}$$
where W—weight and V—volume.

During the November calibration, filtering was performed in the field using predried and preweighed filters in a container. After filtering, the filters were kept, sealed, and labeled in 47 mm Petri dishes.

After analysis for TSS, the filters together with the residues were digested and analyzed to obtain the concentration of total phosphorus on the particles—total particulate phosphorus (TPP)—using the ascorbic acid method [46]. This TPP (termed TPP in TSS, μg mg^−1^) was multiplied by the TSS estimates (mg L^−1^) that were derived from applying the calibration curve to obtain the total particulate phosphorus (TPP in μg L^−1^) at each depth in the water column for every deployment. The equation is shown below:(9)$$\begin{array}{ll} \mathrm{TPP}\;\mathrm{estimate} & \mathrm{in}\;\mathrm{water}\left({\mu g\;L}^{-1}\right)\\ & =\mathrm{TSS}\;\mathrm{estimate}\;\left({\mathrm{mg}\;L}^{-1}\right)*\mathrm{TPP}\;\mathrm{in}\;\mathrm{TSS}\;\left({\mu g\;\mathrm{mg}}^{-1}\right) \end{array}$$

### 2.6. Discharge Data

Instantaneous discharges at UP (25°45′40.8″ N, 80°40′18.8″ W) were downloaded from the S333 inflow structure available at South Florida Water Management District’s online data repository—DBHYDRO (https://www.sfwmd.gov/science-data/dbhydro, accessed on 12 November 2022). Instantaneous discharges at DS (25°45′41″ N, 80°39′17″ W) and INT (25°45′41.0″ N, 80°32′12.0″ W) were downloaded from the United States Geological Survey-National Water Information System (https://waterdata.usgs.gov/nwis, accessed on 26 October 2022). The discharges at INT had both positive and negative values. The negative discharges were taken as flows in the reverse direction (east to west) caused by factors including gate operations at a pump at the eastern end of the canal and wind [47]. Such negative values were converted to positive values for analyses.

## 3. Results

### 3.1. Acoustic Backscatter Processing for the Acoustic Doppler Current Profiler (ADCP)

After converting the RL in counts to EI in decibels (dB) and correcting for beam spreading and water absorption, the relative acoustic backscatter (RB, Equation (3)) from the ADCP was obtained. The RB was not clearly distinguishable between depths in the water column (Figure 3a,b). Using the RB values to develop calibration curves resulted in a very narrow range of backscatter values, for instance, 75.5–77.5 dB at UP (not shown). Applying the equation coefficients (slope and intercept) of the calibration curves to obtain the TSS estimates resulted in values that ranged from 0 to over 1000 mg L^−1^ at UP, which was unrealistic in this relatively low-particulate-level canal (Table 2). Similarly, converting the estimated TSS into TPP resulted in higher concentrations in the surface layer of the water compared to the deeper layer, which conflicted with the measured results that were used for the calibrations. Plots using only the EI yielded better results, with the values being more distinguishable between depths (Figure 3c,d).

### 3.2. Calibration Curves

There were statistically significant positive correlations between echo intensity and log TSS for ADV and ADCP across all three sites, as shown in the calibration curves (Figure 4a,b). Additionally, the uncorrected EI produced better regression for calibration because there was more spread in the EI: 60–70 dB at UP, 50–70 dB at DS, and 55–65 dB at INT (Figure 4b).

### 3.3. Measured Total Suspended Solids (TSS) and Total Particulate Phosphorus (TPP) Used in Calibrations

The gravimetrically analyzed TSS (mg L^−1^) and the resulting calculated TPP in the water column (μg L^−1^) revealed that the top layers of the water column generally had higher concentrations than the deeper layers across the three sites (Table 2). Conversely, the phosphorus fraction of the analyzed sediments (TPP in TSS, μg mg^−1^) generally decreased from the top to the deeper layers. Although the sample collections and analyses were performed at different times of the year across sites, the TPP in TSS was lowest at UP, which was closest to the inflow structure, and highest at INT, which was the most downstream site (Table 2).

During the deployment of the acoustic devices, the discharges at UP had values up to 60 m^3^ s^−1^ (ADV, November) and 40 m^3^ s^−1^ (ADCP, September) (Figure 5a and Figure 6b). During the combined ADV and ADCP deployments at DS in June (DS-Jun), the maximum discharge was approximately 22 m^3^ s^−1^ (Figure 5b), while the discharges during the second ADCP deployment in December (DS-Dec) reached 40 m^3^ s^−1^ (Figure 6f). The DS-Dec deployment had higher discharges because the main inflow structure, S333, and a newer secondary structure, S333N (located north of S333), were both operational (released discharge into the canal) at that time, while during the DS-Jun deployment, only S333N was operational. At INT, the maximum discharge during the combined deployments did not exceed 14 m^3^ s^−1^ (Figure 5c).

In terms of water quality estimates, at UP, the TSS and TPP estimates (concentrations) did not exceed 18 mg L^−1^ and 17 μg L^−1^, respectively, near the canal bed (ADV, Figure 5a) and 15 mg L^−1^ and 13 μg L^−1^, respectively, in the water column (ADCP, Figure 6a and Figure 7a). At DS-Jun, concentrations were less than 14 mg L^−1^ for TSS and 16 μg L^−1^ for TPP near the canal bed (Figure 5b), while in the water column, the concentrations were less than 9 mg L^−1^ for TSS and 17 μg L^−1^ for TPP (Figure 6c and Figure 7b). At DS-Dec, the TSS concentration in the water column peaked at 11 mg L^−1^ (Figure 6e), and the higher concentrations compared to DS-Jun can be attributed to the higher discharges in December. No estimation for TPP in Dec-DS was performed because the number and sizes of measurement depths (ADCP bins) were different between the two deployments. DS-Jun had four 1-m bins, while DS-Dec had five 0.5-m bins, so they were not compatible beyond the calibration curve. Lastly, at INT, the near-bed concentrations did not exceed 5 mg L^−1^ for TSS and 12 μg L^−1^ for TPP (Figure 5c), while in the water column, the concentrations did not exceed 2.5 mg L^−1^ for TSS and 8 μg L^−1^ for TPP (Figure 6g and Figure 7c). These results generally showed higher TSS and TPP concentrations near the canal bed than in the water column.

An observation of the relationship between discharge and the estimates showed clear associations between higher discharges and higher concentrations of TSS and TPP at UP (Figure 5a and Figure 6a,b). The instantaneous and subsequent sustained discharge releases from the S333 structure caused a noticeable increase in TSS and TPP concentrations. Similarly, higher discharges matched higher TSS and TPP concentrations, while lower discharges matched lower TSS and TPP concentrations near the canal bed at DS-Jun (Figure 5b) and for TSS in the water column at DS-Dec (Figure 6e,f). This synchronization was not as strong in the water column at DS-Jun, probably due to fewer particles in the water column compared to the bed (Table 2), while at INT, there was no discernable relationship between discharge and TSS/TPP concentrations due to the relatively lower values of all three parameters.

## 4. Discussion

### 4.1. Justification for Not Correcting for Transmission Losses for ADCP Backscatter Processing

Studies that used ADCPs for TSS estimations and corrected for transmission losses had a wide range of concentrations, relatively high TSS concentrations, and/or greater water depths when compared to this study. For example, in a study performed in the Lembeh Strait, in Indonesia, the measured water depth was between 15 and 30 m, and the measured TSS ranged from 55 to 74 mg L^−1^ [14]. Similarly, in another study performed in the Tidung Island seawaters of Indonesia, the water depth was greater than 40 m, and the TSS ranged from <45 to 80 mg L^−1^ [48]. A study performed in the Hudson River in New York, USA, was conducted in a water depth of 18 m that had TSS concentrations that ranged from <10 and 100 mg L^−1^ [24,49]. Another study performed in San Francisco Bay, USA, had water depths between 7.3 and 16.1 m, and TSS concentrations that were greater than 100 mg L^−1^ [10]. Comparatively, the concentrations in our study were quite low (generally <10 mg L^−1^) and did not vary much between depths, and the water depth in the canal was approximately 5 m. Thus, the poor suitability of the relative backscatter (RB) for developing a useable calibration curve for TSS estimations in this study could be due to the relatively narrow range of acoustic backscatter and TSS in the water column in the L-29 Canal compared to other studies.

The correction introduced for transmission losses increases with increasing distance from the ADCP because the range (distance) from the device is considered. This means that more corrections are applied with increasing distance from the ADCP. Even though, in our study, the bin closest to the ADCP (Bin 1) had the highest RB values that corresponded with the generally higher TSS concentrations at the deeper water column (Table 2), the difference in RB between this bin and the farthest bin was not enough; when the corrections were introduced, the RB values became very similar among all the bins. This resulted in the lack of clear distinction between the RB values with depth, as shown in the results, while the backscatter, uncorrected for transmission losses, produced clearer distinctions with depth. Therefore, the application of such correction “overcorrected” the farthest bins. The use of the Downing near-field factor (Ψ) can overcorrect and bias the relative backscatter, and thus, it may be preferable to use uncorrected data [25], although in some cases, this overcorrection may be marginal [50]. Therefore, it was prudent to remove the two-way correction losses from the relative backscatter processing.

### 4.2. Vertical Profiles of Total Suspended Solids and Total Particulate Phosphorus in the Water Column of the L-29 Canal

The low values of TSS and TPP estimates from this study are not uncommon in the region of Central and South Florida [51,52]. For example, studies conducted in Lake Worth Lagoon (South Florida) and in the tributaries of the Indian River Lagoon (East Central Florida) generally had TSS concentration averages less than 12 mg L^−1^ [52,53] and an average TPP concentration in the water column of less than 70 μg L^−1^ [52]. Similarly, in a study performed along the L-40 Canal, north of the L-29 Canal (our study), the average TSS concentration did not exceed 18 mg L^−1^, and the average TPP concentration was 71 μg L^−1^ [51]. Such low TSS concentrations have been found in <5% of rivers globally [52]. However, the average TPP in TSS from our L-29 study, which was 2.05 μg mg^−1^, is higher than the range of global averages for rivers (1–2.01 μg mg^−1^) [54,55]. Therefore, despite the relatively low TSS in the L-29 Canal, the relatively higher TPP in TSS can threaten the ecological integrity of downstream waters.

The general increase in TSS and TPP estimates from the water surface to the canal bed was expected. We surmise that this is due to resuspension from the bed that caused higher concentrations of TSS [14,56] and the tendency for more suspended sediments to settle out in deeper waters [57]. Therefore, higher suspended sediment levels in deeper waters will result in higher TPP (μg L^−1^) in deeper waters as well. The opposite (decreasing) trend with depth observed for the TPP in TSS (μg mg^−1^) has been reported in other studies (e.g., [56]). A possible explanation for this trend is that the composition of suspended sediments varied from the water surface to the canal bed. The higher TPP in TSS suggests that the water surface had light, organic, and P-rich particles, while the deeper water had denser, more mineral, and P-poor particles. Light and flocculent P-rich organic material (aquatic plant detritus) has been found suspended on the water surface of farm canals [58,59].

### 4.3. Impact of Canal Discharges and Hydrology on Total Suspended Solids and Total Particulate Phosphorus

In southern Florida, canal discharge releases via inflow structures have been shown to increase the concentration and export of suspended sediments and particulate phosphorus in two steps. The first step, referred to as the first flush, marks the start of canal discharge release from the inflow structure after a greater than 24-h quiescent period. This first flush mobilizes and entrains materials deposited from the previous event as well as new material that may have accumulated or grown during the quiescent period. After the first flush is the second step, termed the cumulative flush, where sustained high discharges can mobilize more sediments and their associated P contents [59]. In this study, the effects of first flush and cumulative flush were also observed at the upstream site, closest to the S333 inflow structure, where the increases in TSS and TPP closely followed the discharges from the structure. Instantaneous discharges have been compared to storm events in their capacity to mobilize sediments and increase their concentrations and export [28]. In rivers, the first flush occurs at the start of a rain or storm event, where higher TSS can be recorded during the onset of very high flows that decrease over the duration of the event [52].

Even though the deployments and water quality estimates at the sites were performed at different times during the year, inferences can be made as to the origins and types of suspended sediments in the three canal sites by considering their proximities to the S333 inflow structure. The lower TPP in TSS at UP possibly occurred because high discharges from S333 exported all the fresh, low-density, and P-rich organic particles that accumulated or grew during the quiescent period and left behind the older and denser P-poor particles. Studies have shown that organic particles have a high phosphorus content and are easily transportable, while older particles are more mineral, P-poor, and heavier, which makes them harder to transport [59]. The higher TPP concentrations in the water column at DS suggest that discharges from the secondary inflow structure, S333N, as well as earlier discharges from the main S333 structure, transported P-rich organic particles from upstream which accumulated at the downstream site. Another reason is that the transporting power of the canal discharges was attenuated with increasing distance from the S333 inflow structure; therefore, only the lightest particles made it the farthest downstream. This decrease in transporting power downstream also implies the dominant effect of biogeochemical processes on the kind of particles present and the P availability of such particles. Biogeochemical processes such as adsorption–desorption, biologic uptake, and the decomposition of organic matter have been shown to dominate P availability during low-flow periods [60,61], and in this case, the downstream reaches of the canal can display these low flow conditions. Similarly, the highest TPP in TSS found in the canal interior, at the most downstream site from the S333 inflow structure, indicates that the particles therein may be of biological origin. A high phosphorus mass content has been associated with fresh biological organic matter [59]. These inferences are noteworthy because DS and INT are located close to the bridges that open into the Everglades National Park, and the potential export of P-rich particles into the park will be detrimental to Everglades restoration efforts.

## 5. Conclusions

This study used the acoustic backscatter (echo intensity), a biproduct of velocity measurement from two acoustic devices, an acoustic Doppler velocimeter (ADV) and an acoustic Doppler current profiler (ADCP), to estimate total suspended solids (TSS) and total particulate phosphorus (TPP) in the L-29 Canal in South Florida, USA. The echo intensity from the ADCP, uncorrected for beam spreading and attenuation due to water and sediments, proved sufficient to develop suitable calibration curves to estimate realistic TSS in canal locations that varied by distance from a managed inflow structure, S333. The suspended sediments used in the calibrations were then analyzed for total particulate phosphorus concentrations to obtain the TPP estimates in the water column (ADCP) and near the canal bed (ADV). Time series plots of discharges, TPP and TSS revealed that the site closest to S333, the upstream site, had the highest concentrations, driven by instantaneous and rapid discharge releases. Analyses of the TSS particles further revealed differences between the upstream and downstream sites. Suspended sediments further downstream had a higher phosphorus (P) content per suspended sediments (TPP in TSS μg mg^−1^) because they likely had higher organic matter (biological origin). Conversely, the particles upstream had a lower P content because of their proximity to the high discharges from S333, which transported the lighter P-rich organic particles downstream and left behind denser, P-poor mineral particles. Such important observations and inferences in this study were made possible using high-resolution data from acoustic devices. Therefore, the ability to evaluate how rapid changes in hydrology affect the potential export of sediments and phosphorus is critical for canal discharge management to enable the preservation of the integrity of downstream aquatic systems, such as the Florida Everglades.

This work is novel, because it extends the use of backscatter from acoustic devices, which has been traditionally used as a surrogate for TSS, to also using it as a surrogate for TPP. This is crucial for effective water quality and nutrient (phosphorus) monitoring because it reduces the human and material risks and costs of routine manual samplings and provides high spatiotemporal data. Additionally, this work practically tests the lower limits of using acoustic backscatter as a surrogate for TSS, because the concentrations were at the low end of detection (generally less than 10 mg L^−1^ of TSS), while other studies have focused on much higher concentrations. A general limitation of using acoustic devices as surrogates for TSS and TPP in low-particulate-level water is the possibility of weak signal amplitudes and low echo intensities, which may be indistinguishable from background noise. Another limitation is the potential difficulty in creating low TSS in the laboratory for the calibration of the low echo intensities that may occur under certain field conditions and in certain locations. In many locations, and specifically in the highly managed systems of South Florida, where flow conditions rapidly respond to pumping operations and other water management manipulations, short-term response measurements are needed, as they provide information not obtained by longer-term, less temporally robust data. Future studies should be conducted in several additional canal systems to better determine the range of applicability of these methods.

## Figures and Tables

**Figure 1 sensors-23-02281-f001:**
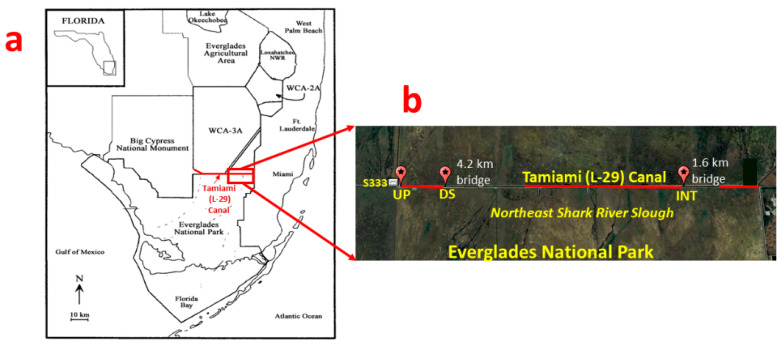
(**a**) Map of the Everglades region showing the Tamiami (L-29) Canal (in red) and (**b**) study sites in the Tamiami (L-29) Canal, where the acoustic devices were deployed. The sites were upstream (UP), downstream (DS), and interior (INT). Modified from Google Earth (2023).

**Figure 2 sensors-23-02281-f002:**
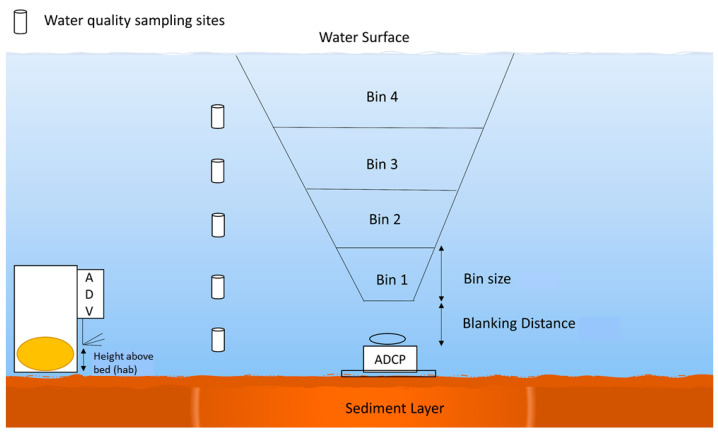
An illustration of a typical deployment of the acoustic devices: acoustic Doppler velocimeter (ADV) and acoustic Doppler current profiler (ADCP). Across deployment events, the user determined the blanking distance and bin sizes of the ADCP ranging from 0.5 to 1 m, and the number of bins varied from 4 to 7. The height above the bed (hab) for the ADV deployments ranged from 0.14 to 0.5 m.

**Figure 3 sensors-23-02281-f003:**
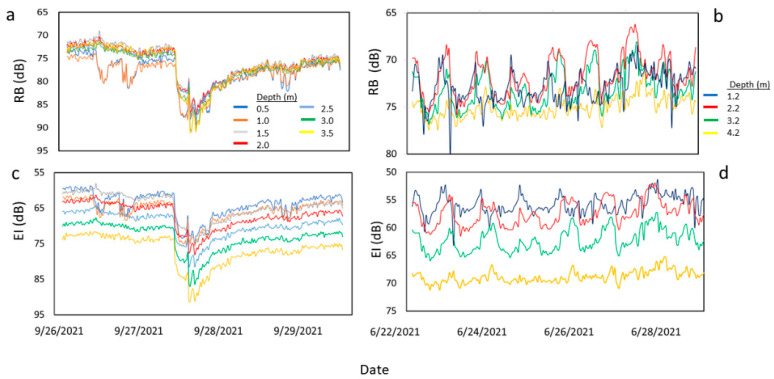
Water column time series of the vertical distribution of the relative acoustic backscatter (RB) (using the average of the beams of each preset bin), corrected for transmission losses, from the acoustic Doppler current profiler (ADCP) at the (**a**) upstream (UP) and (**b**) downstream (DS) sites. Time series of echo intensity (EI) at the UP (**c**) and DS (**d**) show greater differentiation between depths.

**Figure 4 sensors-23-02281-f004:**
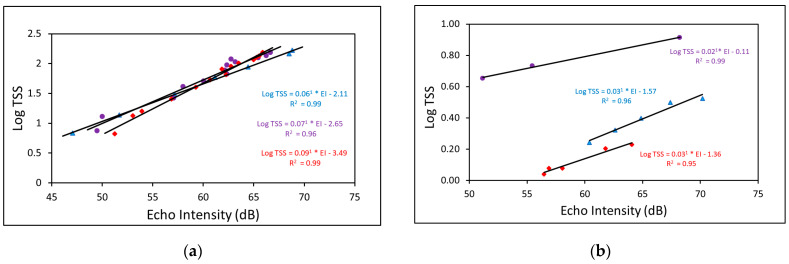
Calibration curves developed using the echo intensity (EI) from the (**a**) ADV and (**b**) ADCP at the upstream (UP), downstream (DS), and interior (INT) sites in the L−29 Canal (for the ADCP, two data pairs/calibration points were removed at UP, while a pair each was removed at DS and IN to improve the fit of the regressions).

**Figure 5 sensors-23-02281-f005:**
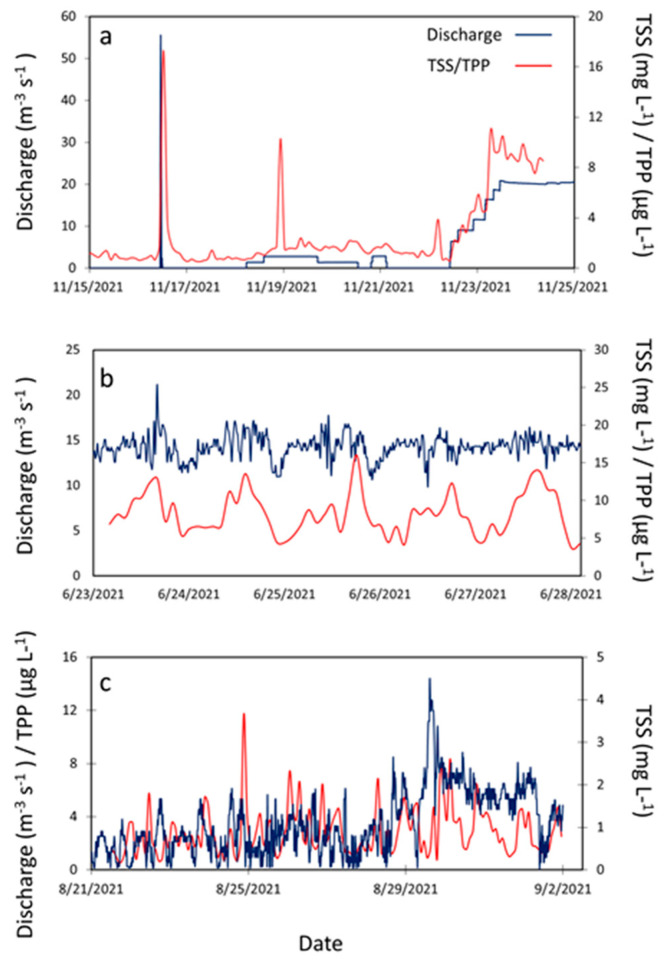
Time series of discharge and estimates of TSS and TPP from the ADV deployments at (**a**) upstream (UP), (**b**) downstream (DS), and (**c**) interior (INT) sites. The TSS and TPP estimates are those near the canal bed. During the field deployments at UP and INT, the measured backscatter obtained from the ADV was lower than the measured backscatter obtained during the lab calibrations because of the inability to artificially create very low TSS concentrations in the lab (<5 mg L^−1^).

**Figure 6 sensors-23-02281-f006:**
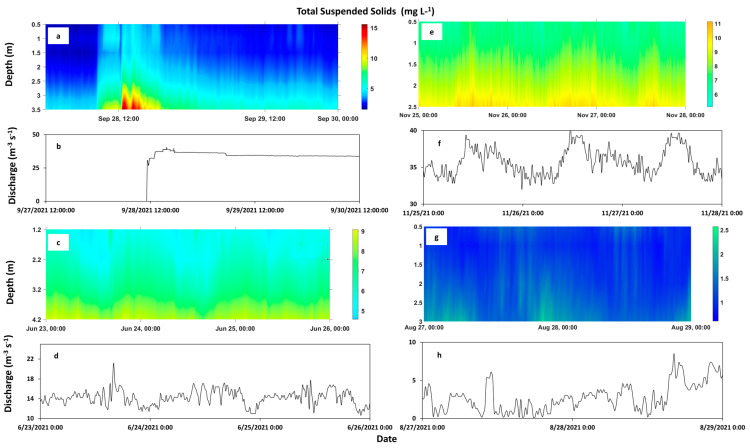
Vertical distribution of estimates of total suspended sediments (TSS) in the water column and the corresponding discharges at the upstream, UP (**a**,**b**), downstream in June, DS−Jun (**c**,**d**), downstream in December, DS−Dec (**e**,**f**), and interior, INT (**g**,**h**), sites in the L−29 Canal in 2021. The depths are the sizes of the user-determined bins of the ADCP. The uncorrected echo intensity (EI) from the acoustic Doppler current profiler (ADCP) was used to generate the TSS estimates via calibration curves. The calibration curve developed for DS−Jun was also used to estimate TSS for DS−Dec. For clarity, these plots are portions of the deployment time series.

**Figure 7 sensors-23-02281-f007:**
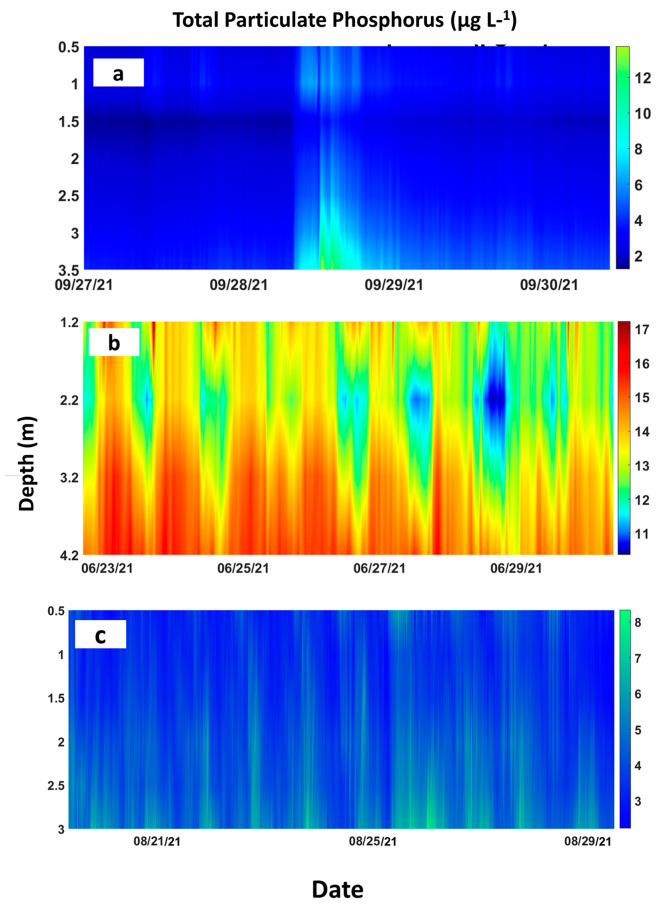
Vertical distribution of total particulate phosphorus estimates (TPP μg L^−1^) in the water column at the (**a**) upstream (UP), (**b**) downstream in June (DS−Jun), and (**c**) interior (INT) sites in the L−29 Canal. The TPP from the lab-analyzed TSS (TPP in TSS) that was used to create the calibration curves (unit of mg g^−1^) was multiplied by the TSS estimated from applying the calibration curves (units of mg L^−1^) to obtain the TPP (unit of μg L^−1^) in the water column.

**Table 1 sensors-23-02281-t001:** Characteristics of acoustic devices manufactured by SONTEK—a Xylem brand.

Parameters	Acoustic Doppler Velocimeters (ADVs)	Argonaut Acoustic Doppler Current Profilers (ADCPs)	References
Sampling volume distance/Max profiling range	5–18 cm	40 m	[32,33]
Max depth range	60–400 m	10–200 m	[33,34]
Min blanking distance	-	0.07 m	[35]
Bin size	-	0.2–30 m	[35]
Max number of bins	-	10 + 1	[33,35]
Pressure sensor	0.1% accuracy	0.1% accuracy	[34,36]
Temperature Resolution, Accuracy	0.01 °C ± 0.1 °C	0.01 °C ± 0.1 °C	[34,36]
Velocity accuracy, resolution	1% of measured velocity, 0.01 cm s^−1^	±1% of measured velocity, 0.1 cm s^−1^	[34,36]
Min and max velocity	0.1 cm s^−1^–5 m s^−1^	<0.01 m s^−1^–6.0 m s^−1^	[32,34,35]
Min Signal to Noise Ratio (SNR) for reliable velocity measurements	5 dB	3 dB	[32,35]
Min correlation coefficient for reliable velocity measurements	≥70%	-	[32]

**Table 2 sensors-23-02281-t002:** Total suspended solids (TSS, gravimetrically determined) and total particulate phosphorus (TPP) content of the TSS (TPP in TSS μg mg^−1^) and in the water column (μg L^−1^), with increasing depth from the water surface of the L-29 Canal. The TSS values were used to calibrate the echo intensity (EI) from the acoustic Doppler velocimeter (ADV) and the acoustic Doppler current profiler (ADCP).

Site	Date	Depth (m) ^1^	TSS (mg L^−1^) *	TPP	
				(μg mg^−1^)	(μg L^−1^)
Upstream (UP)	24 November 2021	0.25	2.10	1.19	2.50
		0.75	1.75	1.27	2.23
		1.25	3.35	0.79	2.66
		1.75	2.50	0.92	2.31
		2.25	3.15	0.91	2.85
		2.75	3.35	0.89	2.97
		3.25	3.25	0.88	2.87
		4.25	2.75	0.95	2.62
Downstream (DS)	22 June 2021	1.2	4.50 *	2.50	11.25
		2.2	5.40	2.23	12.02
		3.2	5.10	2.09	10.64
		4.2	8.20	1.78	14.60
		5.0	26.0	1.15	29.87
Interior (INT)	19 August 2021	0.5	1.20	3.33	3.99
		1.0	1.10	3.66	4.03
		1.5	1.20	3.50	4.20
		2.0	1.30	3.57	4.64
		2.5	1.70	3.26	5.54
		3.0	1.60	3.25	5.19
		3.5	1.80	2.82	5.07

^1^ The lowest depth corresponded to the depth of the ADV and was used to convert the estimated TSS (from calibration) to TPP estimates near the canal bed. * The calibration curve generated using the TSS sampled in June at DS was applied to a second deployment in December.

## Data Availability

Publicly available datasets were analyzed in this study. These data can be found here: https://www.sfwmd.gov/science-data/dbhydro and https://waterdata.usgs.gov/nwis (accessed on 1 November 2022).

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
