# Peer review of "High-Resolution Estimation of Suspended Solids and Particulate Phosphorus Using Acoustic Devices in a Hydrologically Managed Canal in South Florida, USA"

_sensors, 2023, doi:10.3390/s23042281_

Round 1

Reviewer 1 Report

The article is well written, scientifically argued and represents an important contribution in the field.

Author Response

Thank you for the review of our manuscript

Reviewer 2 Report

Dear Authors,

Thank you for the article you sent, which undoubtedly raises important aspects of water quality in flowing waters.

However, please consider a few of my comments:

- conclusions are too general. Please compare your results with other authors.

- please clearly state in the applications what are the limitations in the application of the proposed methodology.

- are further studies planned in other channels?

- Fig 1 a) please change it, you can't see anything on it.

- Fig 3 describe the w axes in the missing graphs.

- for each pattern, please provide the following explanations:

e.g. formula (2)

LogTS = A+B*RB

Where:

A-.....

B- .....

RB - .....

The current format is illegible, the text blends. Please clearly explain the factors in all formulas.

Kind regards.

Author Response

Thank you for your insightful feedback.

  • In response to the comment of our "general" conclusions we have expanded on the conclusions, and highlighted the novelty of this work.

(e.g., L550 - 568) This work is novel, because it extends the use of backscatter from acoustic devices, which has been traditionally used as a surrogate for TSS, to also using it as a surrogate for TPP. This is crucial for effective water quality and nutrient (phosphorus) monitoring because it reduces the human and material risks and costs of routine manual samplings and provides high spatiotemporal data. Additionally, this work practically tests the lower limits of using acoustic backscatter as a surrogate for TSS, because the concentrations were at the low end of detection (generally less than 10 mg L-1 of TSS), while other studies have focused on much higher concentrations. A general limitation of using acoustic devices as surrogates for TSS and TPP in low-particulate-level water is the possibility of weak signal amplitudes and low echo intensities, which may be indistinguishable from background noise. Another limitation is the potential difficulty in creating low TSS in the laboratory for the calibration of the low echo intensities that may occur under certain field conditions and in certain locations. In many locations, and specifically in the highly managed systems of South Florida, where flow conditions rapidly respond to pumping operations and other water management manipulations, short-term response measurements are needed, as they provide information not obtained by longer-term, less temporally robust data. Future studies should be conducted in several additional canal systems to better determine the range of applicability of these methods.   

  • The limitations are addressed in the methods and in the conclusion

  Methods:

  • Table 1
  • 1.1 Characteristics and limitations of acoustic devices

(e.g., L94 - L115) Acoustic devices collect many types of data including velocity, signal amplitude/signal-to-noise ratio (SNR), correlation coefficient, temperature, and pressure (Table 1). Velocity is typically the primary parameter of interest, while the signal amplitude and correlation are used to provide data quality information [31]. The main purpose of the signal amplitude is to determine if there are enough particles in the water, making it an excellent illustration of sediment dynamics and fluctuations [32]. This signal amplitude constitutes the acoustic backscatter parameter used as a surrogate for total suspended solids. The signal amplitude data are usually accessed from the device as either signal amplitude (units of counts) or the SNR. For SONTEK instruments, the SNR is obtained from the signal amplitude by subtracting the ambient, background electronic noise level and converting it to decibel units (dB) (multiplying by 0.43) [32]. To obtain accurate velocity measurements, SNR values must be at least 4 dB (ADV) and 3 dB (ADCP) (Table 1). An SNR value of 0 dB indicates that the water is too clear and that there is no distinction between the signal (acoustic energy) and the ambient noise level [31,32]. The ADV uses the correlation coefficient as a second data quality parameter, and it is expressed as a percentage. A correlation score of 100% means reliable low-noise velocity measurements, which occur at an SNR level of about 15 dB, while a 0% correlation reflects velocity measurements dominated by noise [31,32]. Interferences that can lead to low correlation values include high velocities, turbulence, and aerated water. A score of 70% or higher indicates accurate velocity data [32].

In response to future studies and method applicability we wrote a concluding statement (additionally we are seeking funding for additional studies in other areas but are, as of yet, in the pre-proposal stage).

In many locations, and specifically in the highly managed systems of South Florida, where flow conditions rapidly respond to pumping operations and other water management manipulations, short-term response measurements are needed, as they provide information not obtained by longer-term, less temporally robust data. Future studies should be conducted in several additional canal systems to better determine the range of applicability of these methods.     

  • Figures 1 and 3 have been changed
  • Formulae have been outlined and explained in the equation boxes, and the font size has been increased

 The Language "English" grammar has been edited by use of MDPI's language service.

Reviewer 3 Report

I have some minor comments and suggestions to the authors:

1. The analytical characteristics of a new sensor require information about limits of detection and determination of the analytes; linear range of the calibration curve; possible interfering agents (of chemical or physical nature); budget of uncertainty; sensitivity of the method (sensor); reproducibility; time stability of the signal.

2. A Table with data about these parameters will be of use for possible users and will help for optimization studies.

Author Response

  • See for example L96-114 and Table 1.
  • The limitations are addressed in the methods and in the conclusion

  Methods:

  • Table 1
  • 1.1 Characteristics and limitations of acoustic devices

Acoustic devices collect many types of data including velocity, signal amplitude/signal-to-noise ratio (SNR), correlation coefficient, temperature, and pressure (Table 1). Velocity is typically the primary parameter of interest, while the signal amplitude and correlation are used to provide data quality information [31]. The main purpose of the signal amplitude is to determine if there are enough particles in the water, making it an excellent illustration of sediment dynamics and fluctuations [32]. This signal amplitude constitutes the acoustic backscatter parameter used as a surrogate for total suspended solids. The signal amplitude data are usually accessed from the device as either signal amplitude (units of counts) or the SNR. For SONTEK instruments, the SNR is obtained from the signal amplitude by subtracting the ambient, background electronic noise level and converting it to decibel units (dB) (multiplying by 0.43) [32]. To obtain accurate velocity measurements, SNR values must be at least 4 dB (ADV) and 3 dB (ADCP) (Table 1). An SNR value of 0 dB indicates that the water is too clear and that there is no distinction between the signal (acoustic energy) and the ambient noise level [31,32]. The ADV uses the correlation coefficient as a second data quality parameter, and it is expressed as a percentage. A correlation score of 100% means reliable low-noise velocity measurements, which occur at an SNR level of about 15 dB, while a 0% correlation reflects velocity measurements dominated by noise [31,32]. Interferences that can lead to low correlation values include high velocities, turbulence, and aerated water. A score of 70% or higher indicates accurate velocity data [32].